# Seismic Vulnerability Assessment of a Medieval Urban Cluster Identified as a Complex Historical Palace: Palagio di Parte Guelfa in Florence

Anna Caranti [1], Vieri Cardinali [2], Anna Livia Ciuffreda [3], Massimo Coli [3], Mario De Stefano [2], Elena Le Pera [1] and Marco Tanganelli [2],*

1 Independent Researcher, 50121 Florence, Italy
2 Department of Architecture, University of Florence, 50121 Florence, Italy
3 Department of Earth's Science, University of Florence, 50121 Florence, Italy
* Correspondence: marco.tanganelli@unifi.it

**Abstract:** This paper presents the results of the application of a holistic procedure for a seismic vulnerability analysis of complex masonry aggregates which are ascribable as cultural heritage buildings. The methodological framework allowed us to properly identify the structural units comprising a historical palace through a hierarchical approach combining integrated geometrical and structural surveys with non-destructive (ND) and minor destructive (MD) techniques. The investigations were conducted on a significant palace located in the historical center of Florence, the Palagio di Parte Guelfa. The building covers an entire urban block, as it is constituted by many structural units developed over the centuries since the Roman period. The palace incorporates pre-existing structures, such as the Church of Santa Maria Sopra Porta and medieval terraced houses. Over the decades, different restorations and renovations have been carried out, including by Filippo Brunelleschi in the XV century and by Vasari in the XVI. Nowadays, the building constitutes an urban cluster. Our seismic vulnerability analysis took advantage of a specific knowledge path which was suitable for the identification of the different structural units of the palace. To this end, the historical evolution of the construction was traced through bibliographic research and ND surveys. We finally assessed the seismic performance of the different units according to different approaches, i.e., a rapid assessment based on simplified computations at the global scale, and a kinematic analysis for local phenomena.

**Keywords:** cultural heritage; seismic assessment; medieval urban cluster; complex buildings; masonry structures

## 1. Introduction

The structural assessment of historical buildings is of tremendous significance. Historical constructions need to be preserved in order to transmit their heritage to the following generations [1]. In this regard, earthquakes are among the most violent events threatening the safeguard of such constructions [2]. Several cases in southern European, and in many other parts of the world, have shown the fragility of built environments when faced with ground motions [3–5]. Hence, much effort has been made by the scientific community to find robust approaches to preserve non-engineered and vernacular structures [6–8]. Besides the standard constructions that characterize the historical centers of cities around the world, specific attention must be paid to cultural heritage buildings (CHBs) [9]. Several studies have been conducted in recent years in attempts to devise reliable numerical approaches for seismic assessments of monuments [6,10,11]. Among them, the PERPETUATE project developed specific guidelines for performance-based assessments of CH structures [12]. The significance of CHBs has been also evaluated on wider scales. Recently, the authors

of [13] presented a comprehensive evaluation of the seismic risk to UNESCO Cultural Heritage sites in Europe.

In Italy, the "Guidelines for the evaluation and reduction of seismic risk of cultural heritage" [14] indicate three levels of evaluation (LV1, LV2 and LV3). The first level (LV1) represents a simplified approach to define a risk ranking of different CH structures within the Italian territory. This classification has been validated by the use of more refined or simplified approaches [15,16], and it can be adopted to assess scenarios at different levels, from the city scale to wider territorial perspectives [17–19]. The last two levels of evaluation specifically refer to single structural units (SU). LV2 involves the local mechanisms of the macro-elements constituting the investigated structures. This assumption relies on the hypothesis that, in absence of a global box-behavior, masonry structures tend to respond to seismic actions through local phenomena [20–22]. Kinematic analysis is an efficient tool to predict the seismic behavior of structures at low computational cost. In recent years, several advances have been made [23–27]. Finally, the third level of assessment, LV3, indicates global seismic assessments of masonry buildings. Besides the different modelling approaches available [11], such assessments require global behavior under seismic motion [21,28].

Beyond the methodologies for structural assessment, in the seismic vulnerability analyses of CHBs, the identification of the structural behavior and the determination of the mechanical properties of materials are crucial. In this regard, national and international codes require the development of specific cognitive paths [14,29]. The latter seek to understand the historical evolution of buildings in order to identify structural units to be evaluated. In this context, multidisciplinary research and expertise are required [30–35]. A relevant issue is covered by the in situ diagnostic campaigns [36–38], which can be divided into destructive (DT), minor destructive (MDT) and non-destructive techniques (NDT). While the first two approaches are preferred, their extensive application on CHBs with artistic value is discouraged. Therefore, MD and NDT are now widely used to characterize the structural features of CHBs [39]. Nevertheless, all the different tests have their shortcomings in terms of comprehending the historical evolution and the alterations that characterize the investigated structures [40,41]. If the historical evolution is well documented and the modification is traceable, it will be easier to perform effective experimental tests. However, the historical centers of European cities date to hundreds/thousands of years ago. Hence, the alterations and transformations that have occurred are not always traceable, when historical documentation is not available. In these cases, experimental tests can provide new information regarding structural features [42]. In situ campaigns can be planned based on the priority of each of the different tests. In [34], a funnel-shaped procedure for the structural characterization of CHBs defined dependent relationships between DT and MD tests. Namely, DT tests are extensively adopted to first characterize the investigated structure; then, MD tests are used on certain identified structural parts to validate the first assumptions and qualitatively characterize the applied technologies.

The present work performs a seismic vulnerability assessment of a complex palace occupying an entire urban aggregate. Seismic assessments of heterogenous and complex cultural heritage buildings involve several issues, i.e., the evolution of the structure, its effect inside the aggregate system, and comprehension of the structural behavior of the different parts [43–47]. The seismic assessment of the Palagio di Parte Guelfa required an understanding of the structural units of an entire urban block. In this work, a methodological framework combining historical research and integrated architectural and structural surveys with ND and MD techniques has been adopted. We show the feasibility of the procedure to investigate complex CHBs. The research allowed us to define the structural units of the Palagio di Parte Guelfa, which could be divided into six major structures. The ND and MD tests, executed following a hierarchical logic, provided information on the quality of the technologies and materials adopted. Finally, seismic analyses were executed according to two different approaches: a simplified one, intended to define a global safety

index of each structure, and a kinematic analysis, which was found to be the most reliable approach due to the lack of connections between the different structural parts.

## 2. Palagio di Parte Guelfa in Florence

### 2.1. The Knowledge Path

Palagio di Parte Guelfa is located inside the historical center of Florence. Due to its aggregations and historical evolution, it can be considered one of the most important structures representing the medieval city of Florence. The building, which took shape during the XIII century, represented the core of the politic scene during the XIV and XV centuries. The structure has had important renovations up to the XX century. In Figure 1, the layout of the Palagio within the Florentine urban structure is presented. From the roofs of the building, it is possible to observe the numerous differences in terms of the various levels and structures that characterize the palace.

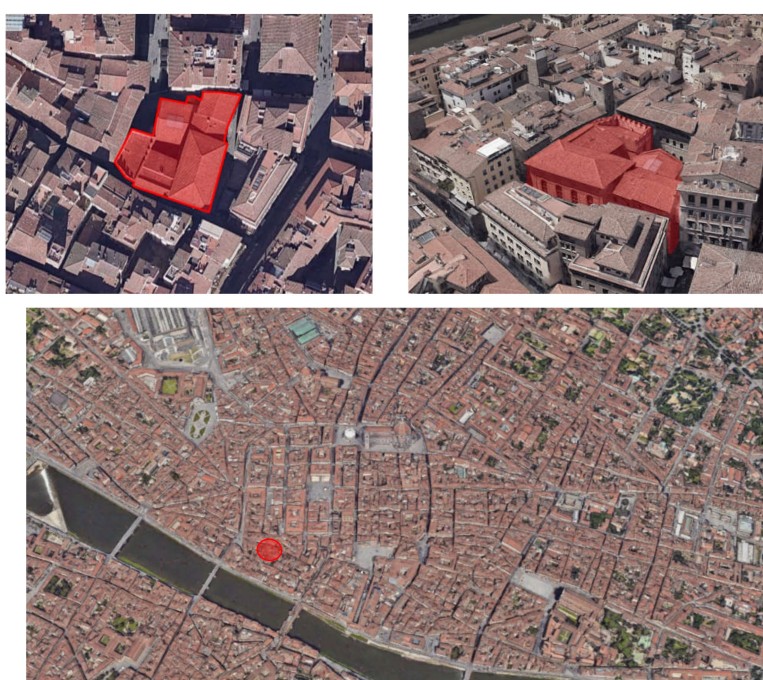

**Figure 1.** Location of the palace in Florence; the building is highlighted in red (source Google maps, Google Earth).

The seismic vulnerability assessment of the complex followed the methodological flowchart shown in Figure 2. The procedure can be divided into several substeps. Preliminary investigations were conducted, comprising in situ inspections of the dimensions of the palace in order to determine its relationship within the adjacent units. This phase anticipated the execution of geometrical surveys or the application of diagnostic techniques. During such preparatory evaluations, preliminary structural models can be realized on the basis of the available information. These can be useful to execute sensitivity analyses, highlighting uncertain parameters that are worthy of further investigation [48]. In this work, before our investigation of the building, historical research was undertaken, based on chronicles published by different authors. The combination of the outcomes of historical studies and preliminary in situ observations allowed us to devise a diagnostic campaign. In this research, all the investigations shown in Figure 2 were carried out. Namely, architectural and structural surveys of the building were executed. Then, several ND and MD techniques were performed, such as thermography tests, GPR surveys, sonic tests on the masonry walls, and drilling inspections. All of the proposed tests were ranked according to their priority: (i) ND tests were extensively used to investigate the different parts of the Palace; and (ii) MD tests were undertaken in order check the reliability of the ND tests

and qualitatively characterize specific parts of the structure. On these bases, an architectural survey of the Palagio was performed. The geometrical parameters derived from the architectural digital survey, i.e., combining the outcomes of the executed diagnostic campaigns, were later characterized. Finally, the acquired knowledge allowed us to identify the structural units of the palace and to characterize its materials and their properties for our subsequent seismic analyses. The procedure followed the assessment approach for CHBs provided in [14]. First, a simplified LV1 method was used for all models according to the most reliable approach based on the structural features of the units. Then, the different structures were investigated based on the expected behavior under seismic motion in order to determine whether the units could exhibit a global behaviour or they were suitable to be investigated by kinematic analyses.

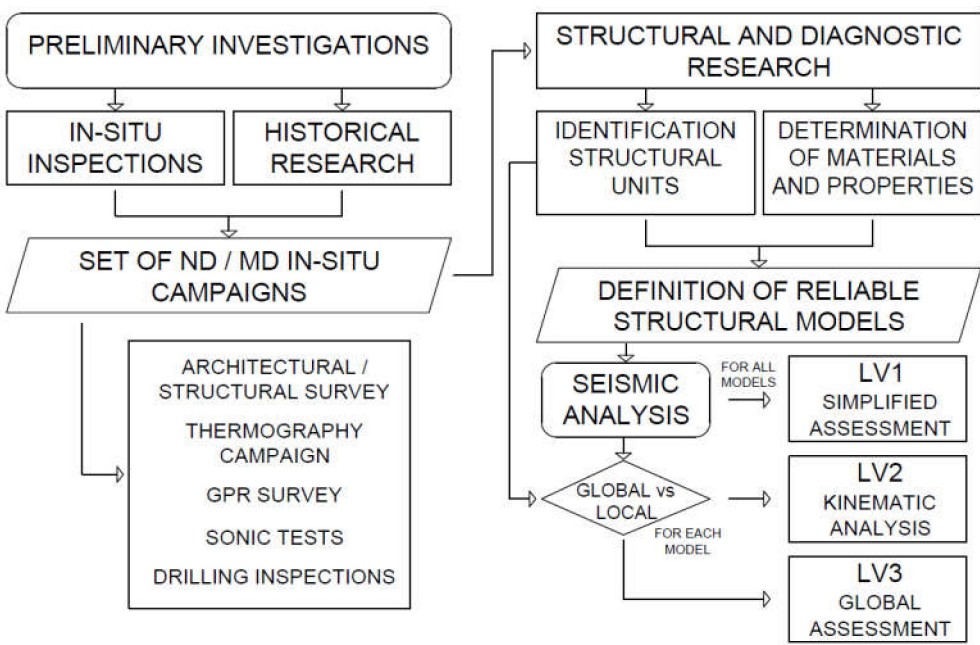

**Figure 2.** Methodological flow-chart of the research.

2.1.1. Historical Evolution

Historical research was conducted to evaluate the available bibliographical sources with relevant information about the Palace. Specifically, in this section, we take advantage of research carried out by [49–56]. Palagio di Parte Guelfa is located in an area which has been urbanized since the Roman period; therefore, the historical evolution of the building is connected with the presence of other activities and buildings. Specifically, the Parte Guelfa, the Santa Maria Sopra Porta Church and the Arte della Seta represented the three institutions involved in the medieval transformations of the area. The original layout of the Palagio dates from the XII century, when the Guelfs, powerful in the city during the medieval period, were looking for new headquarters. They bought a portion of the complex close to the Santa Maria Sopra Porta Church. Historical information concerning the church is scarce; the available documentation refers to the enlargement of the structure for the realization of a new chapel in 1345 (Figure 3).

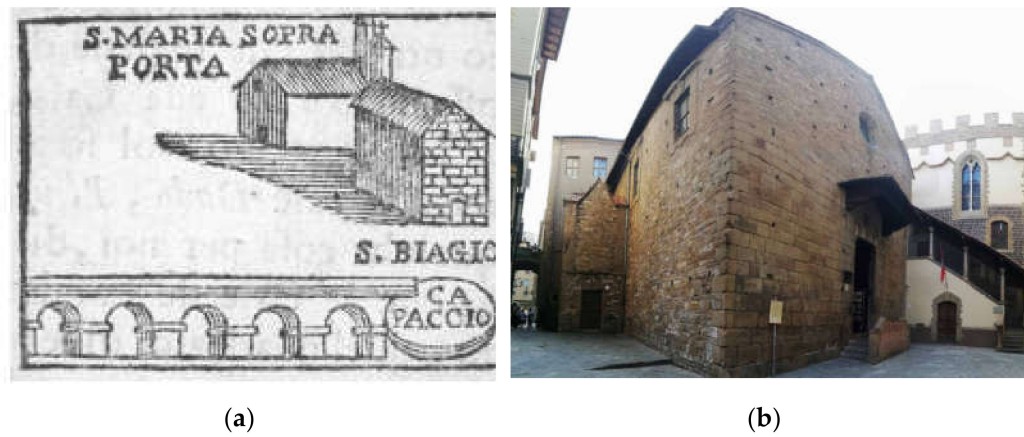

**(a)** **(b)**

**Figure 3.** The Santa Maria Sopra Porta Church: (**a**) historical documentation from [52]; (**b**) photo of the main entrance.

Later, the ecclesiastic building suffered from an economic crisis and the new Chapel was purchased by Parte Guelfa in 1410. The Parte Guelfa, i.e., the Guelph Party, was in opposition to the Ghibellines. Historically, the two factions supported the Pope and the Holy Roman Emperor, respectively. The first Palagio di Parte Guelfa comprised a big single room located on the first floor, while the ground floor was for commercial use. In the same area, other structures existed in the medieval period. The Arte della Seta, the Silk Guild of Florence, established its headquarter in an adjacent structure located on a corner close to the Mercato Nuovo, the Renaissance Market of the City. After this operation, the Arte della Seta commissioned several renovations to expand the building, i.e., Udienza's room in 1385 and the Audienzetta in 1422. In the XIV century, the Audienzetta was further expanded by the Parte Guelfa, as evidenced by documentation for the obtainment of properties facing Via delle Terme. In the XV century, two main interventions can be highlighted. The first regarded the nucleus facing Via delle Terme, while the second one involved the Sala Grande room, located along Via di Capaccio (Figure 4). The rise of the Medici family in the XV century led to a slow decline of the power of the Parte Guelfa. In this context, the Guelphs commissioned an extension of the palace in the form of the erection of a new Council Hall. The design for the new Sala Grande is attributed to Filippo Brunelleschi. Although the presence of the architect on site is not documented, he provided indications for the execution of the construction. Nevertheless, he did not follow the project continuously, and it seems that the Guelphs had to conclude the work by calling local craftsmanship and betraying the original layout of Brunelleschi. Vasari planned other interventions in 1558, after the flood of 1557; however, documentation on this topic is scarce, and it is not possible to identify the specific alterations that were made. Besides the church, at the time, the ground floor of the palace comprised private workshops, while the upper levels were occupied by two institutions: the Parte Guelfa and the Monte Comune, i.e., the office managing loans of citizens to the municipality of Florence. Other interventions followed during the succeeding centuries. In the XIX century, the local fire brigade was based in the structure, leading to further alterations and the demolition of some of the original structures. In the first decades of the XX century, architect Alfredo Lensi carried out a new restoration project. Several vaults were refurbished, and many rooms were restored at their original configurations. In 1944, the building was identified as one of the structures to be preserved during WWII. Nonetheless, the palace suffered considerable damage due to debris from nearby structures which had been mined by the retreating Nazis. In 1966, the flood of Florence hit the structure and other restorations followed. Further restorations occurred because of water leaks in the roofs. Nowadays, the Palagio di Parte Guelfa is used for several purposes. A public library is located inside the old church, while the ground floor on Via delle Terme is used by the traffic wardens of the City of Florence. On the upper levels, rooms for events and meetings are available, while other spaces are occupied by

different institutions and public offices. In Figure 5, a representation of the evolution of the building, showing the major interventions since its origin, is shown.

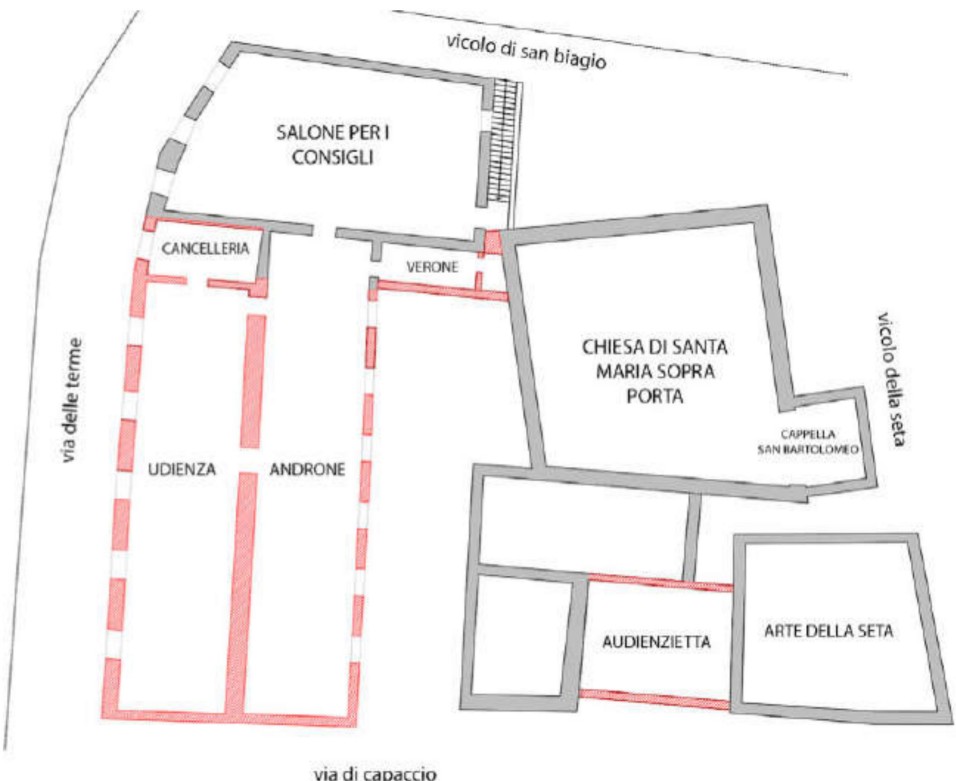

**Figure 4.** Plan of the first floor: hypothesis of the transformations in the years 1415–1426. In grey, the existing structures at the time are reported. In red, possible new additions are highlighted.

### 2.1.2. Architectural and Structural Survey

A structural survey of the Palagio was conducted in the form of an integrated remote sensing survey. This was executed using a Faro Focus S 300 laser scanner from the Department of Architecture of Florence. Based on the research conducted on the historical evolution of the Palace, together with the crack survey and the execution of ND campaigns, we were able to identify the different structural units present. In Figure 6, a characterization of the ground floor level and a 3D view of the palace is presented. Our survey of the crack patterns did not reveal significant problems related to ongoing movements or other deficiencies, showing only minor damages which are typical for old masonry structures.

### 2.1.3. Thermography Campaign and GPR Survey

In this evaluation, only ND campaigns were executed, i.e., investigations that could potentially damage the building were not carried out. Especially when combined, ND techniques can provide satisfactory qualitative results [57,58]. In this work, ND tests were conducted extensively throughout the palace.

Infra-red thermography campaigns and GPR surveys were executed extensively throughout the different structural units. For the thermography, a FLIR T460 camera was used, while for the GPR, a C-Thrue antenna (by IDS Georadar S.R.L. Hexagon Group) was used. The two ND systems provided different information [59–61]; for example, infrared thermography can detect temperature variations inside the various elements comprising the structure [62], while GPR surveys can identify alterations hidden under the plaster layer, such as pre-existing arched windows or heterogeneous masonry textures in walls [63]. In other studies, connections within orthogonal walls have been identified using thermography. At the same time, GPR allows the collection of other qualitative information regarding both masonry walls and horizontal elements [64–67]. In our study,

GPR surveys of the horizontal elements were undertaken, starting from the extrados of slabs and vaults. In different cases, such as for the Arte della Seta unit, we identified a reinforced concrete slab over the existing wooden slabs. The combined used of the two ND techniques allowed us to identify different membranes, which are important to define the stiffness of the diaphragms and their capacity to absorb seismic actions. Different modern slabs (such as SAP slabs) were found on the ground floor along Via delle Terme. In other cases, the GPR surveys ruled out the presence of reinforcements over the vaults.

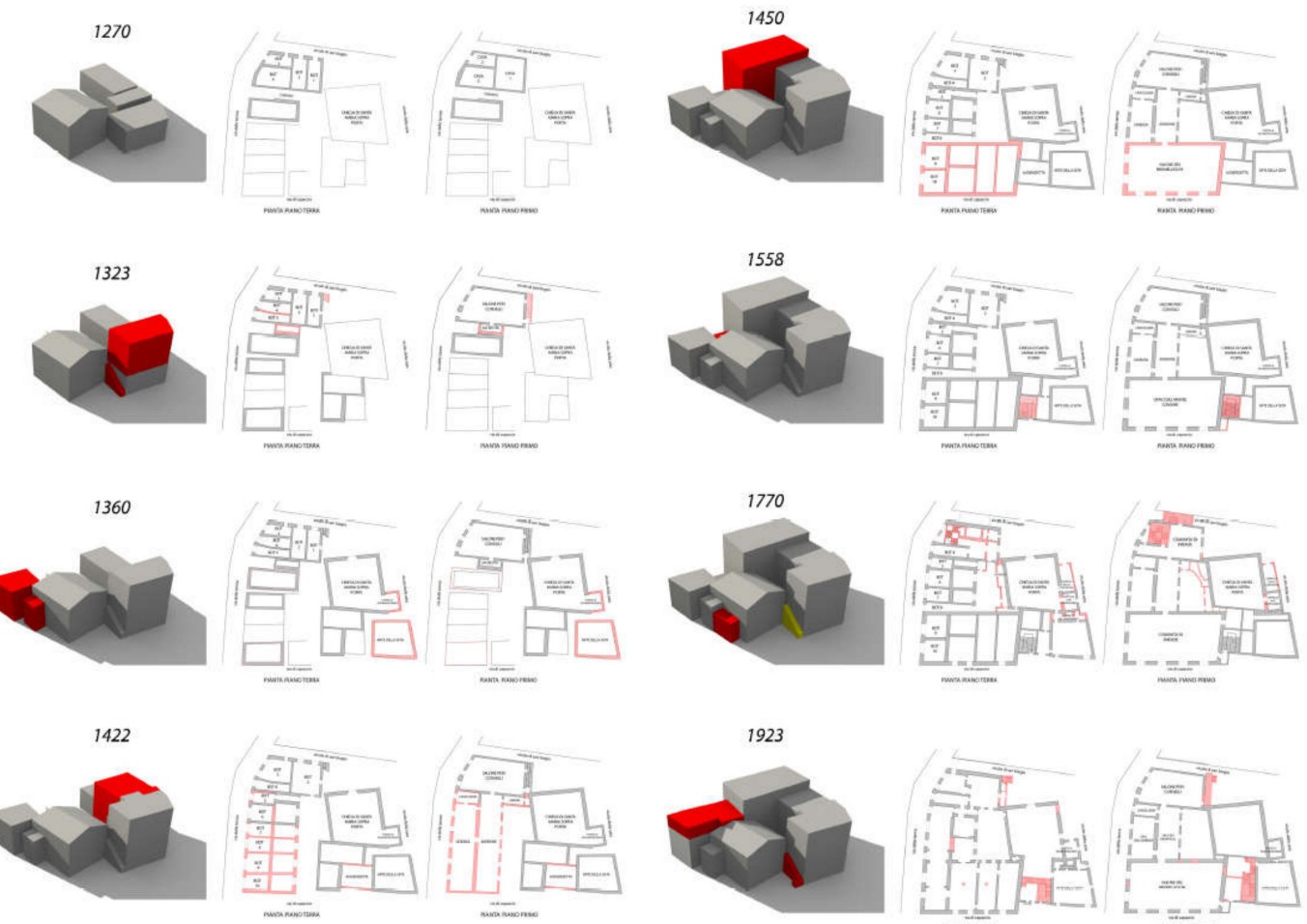

**Figure 5.** Summary of the transformations which occurred from 1270 to 1923. For each phase, the red parts indicate the spaces constructed during those years.

Concerning the masonry walls, the two systems combined allowed us to identify three main masonry typologies:

− Compact and homogenous mixed masonry
− Non-homogenous mixed masonry characterized by the presence of cavities and rubble stones with variable dimensions
− Masonry with double leaves and an inner nucleus

Our analysis of the outcomes of the campaign showed that the first type of masonry was mainly present on the lower levels, while the other two typologies were used on the upper ones.

Palagio di Parte Guelfa—structural units

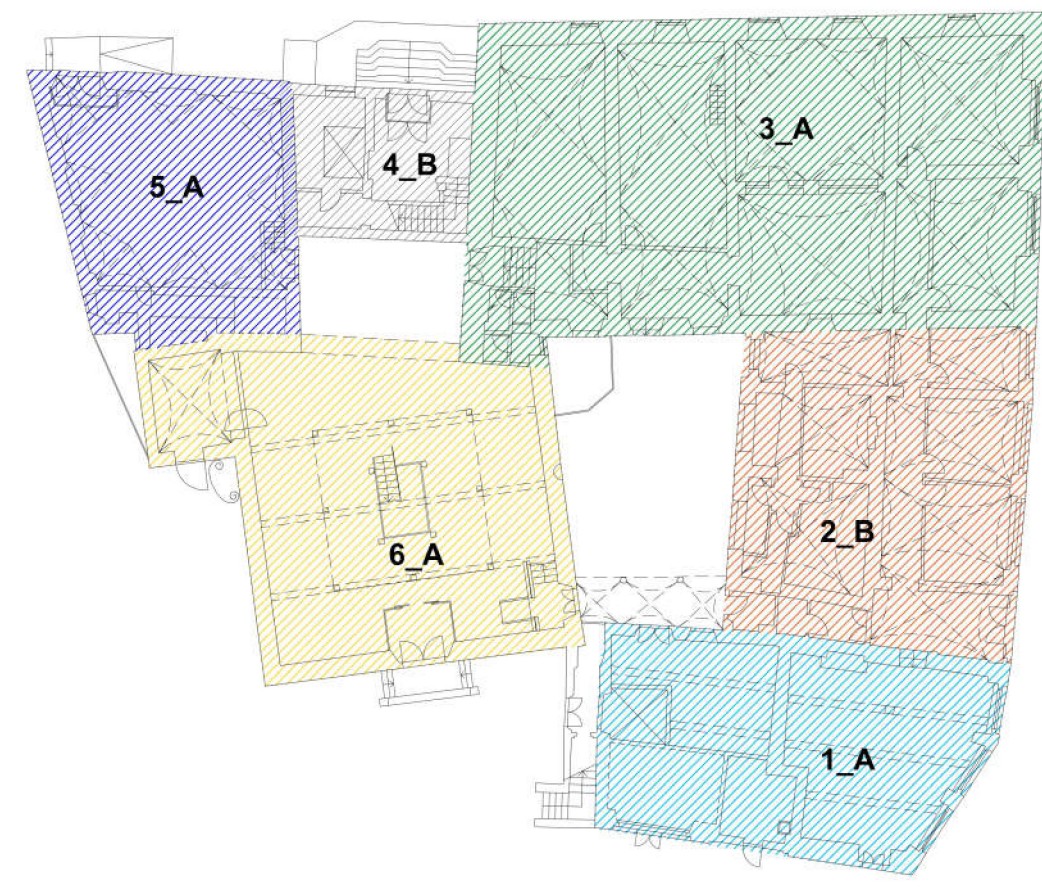

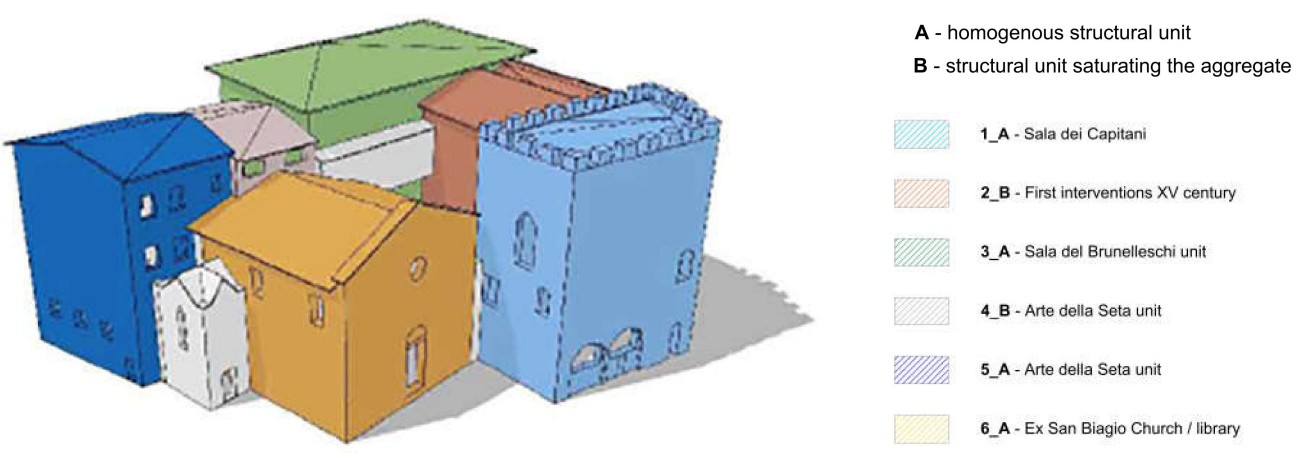

**Figure 6.** Identification of the structural units. The letter A indicates homogenous units, while B denotes units made to saturate the spaces. 1_A—Sala dei Capitani (XIV century); 1_B—First intervention in the XV century; 3_A—Sala di Brunelleschi; 4_B—Arte della Seta annex; 5_A—Arte della Seta main unit; 6_A—Ex San Biagio Church.

2.1.4. Sonic Tests and MD Campaigns

The results of the tests conducted in the two ND campaigns were verified through two other types of investigation: sonic tests and drilling tests for inspections using endoscopy. The sonic tests executed over the masonry walls allowed us to perform an additional qualitative evaluation of the masonry typologies, determining their dynamic elastic moduli. The drilling test and endoscopy allowed us to check the initial results concerning the inner parts of the masonry.

The sonic tests were executed with a Novasonic U5200 CSD, comprising a load cell connected with a hammer and an acquisition system which is able to measure impacts. As the wave propagation is influenced by the geometry of the section and the material properties of the layers, we could determine the velocity of transmission of the signal, and with this, the dynamic modulus of the material [68,69]. The results of these tests were classified according to the wave velocity of transmissions of the masonry panels, identifying three distinct categories: velocities lower than 1000 m/s, between 1000 and 2000 m/s, and greater than 2000 m/s [69,70]. The first category indicates panels with inner cavities and discontinuities, probably due to non-homogenous nuclei or detachments within the external leaves and the nucleus. The second category, to which most of the panels belong, includes masonry typologies characterized by good texture and minor cavities in the inner parts. Finally, well-realized masonry panels with good material properties constitute the third category.

Forty-four endoscopic tests were executed to check the accuracy of the GPR and sonic tests through the direct visualization of the internal parts of the walls. This was made realizing our MD campaign, were we executed drilling holes with a diameter of 16 mm inside the thickness of the walls. In Figure 7, the building layout summarizing the different tests conducted is presented.

*2.2. Identification of the Structural Units*

The structural units of the palace were investigated through four different ND techniques that, combined with the evidence of the architectural survey and the historical research, allowed us to determine the structural features of the building. From a vulnerability perspective, the masonry typologies defined initially after the GPR survey are associated with two current classifications described in the literature, i.e., one provided by the Tuscany Region [71] and the other from the Italian technical code [29] for the whole Italian territory. In Table 1, the two distinct nomenclatures are shown. It is worth noting that the classifications provided in the regional list do not entirely align with those outlined in [29]. In other cases, e.g., in modern masonry typologies, the Italian code lacks a specific classification for typologies characterized by resistant elements with block dimensions. In Table 2, the masonry typologies of the Palagio di Parte Guelfa are listed using both classification systems. In the case of masonry types B and C, the same MIT type, i.e., rubble stone masonry, applies. Further classifications of the different chaotic stone masonries used in medieval buildings in the Tuscany area can be found in [72]. In Figure 8, the layout of the building is presented. The different identified masonry typologies are attributed to the different walls; the dashed walls indicate the suppositions made according to the NT campaigns, while solid lines indicate panels verified with the endoscopy visualization.

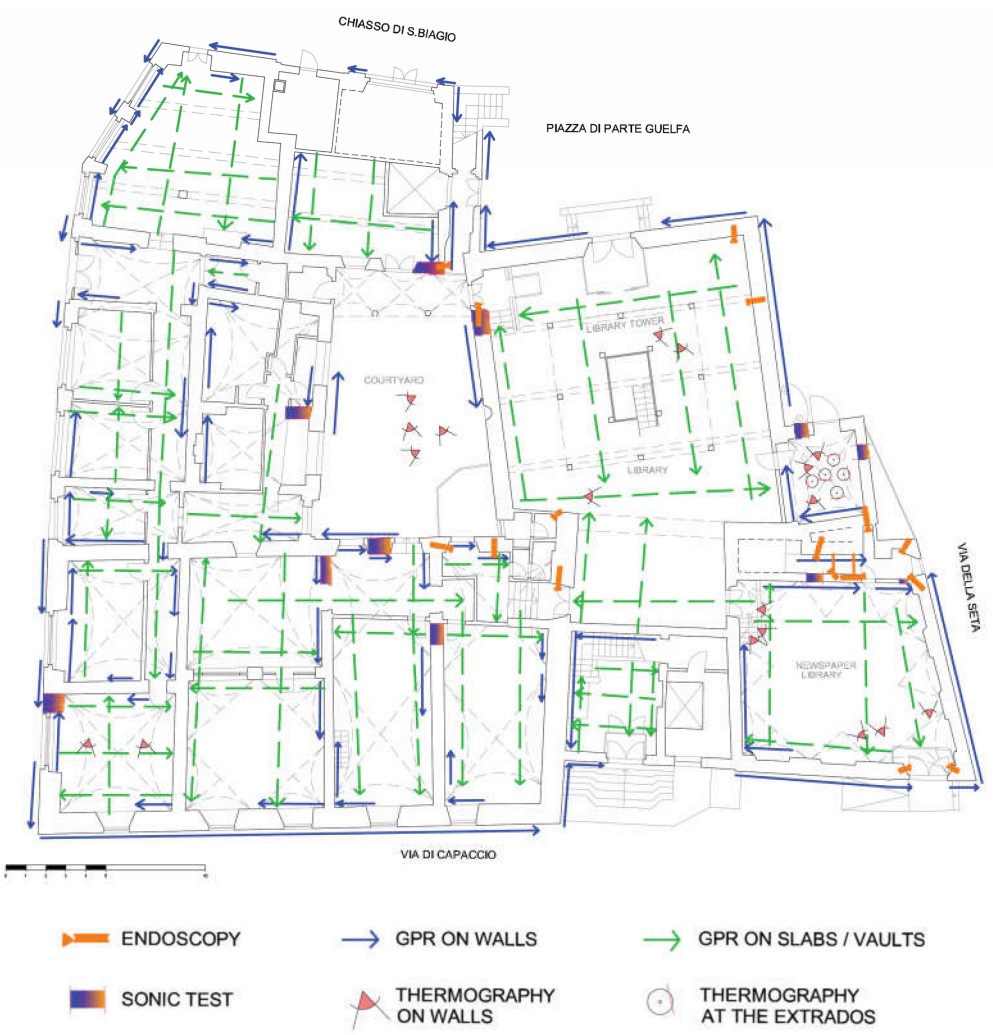

**Figure 7.** The ground floor plan of the building with the investigations carried out. Other layout diagrams are included in Appendix A.

**Table 1.** Masonry typologies according to the regional classification (left) and the national one (right).

| | Tuscany Region [71] | | Italian Code [29] |
|---|---|---|---|
| A | Masonry comprising an inner nucleus and two external leaves realized with variable stones, poorly disposed and without a connection between the two sides of the panel | 1 | Rubble stone |
| B | Masonry comprising an inner nucleus and two external leaves realized with stones of regular dimensions, well-disposed and with connections between the two sides of the panel, horizontal layers of chiselled stones or clay bricks | 2 | Rough blocks with non-homogeneous thickness of the external faces |
| C | Rough stone masonry with irregularities | 3 | Split stone with good textural disposition |
| D | Rough stone masonry with horizontal layers of chiselled stones or clay bricks | 4 | Irregular soft stone (tuff, calcarenite etc.) |
| E | Rubble stone masonry of variable dimensions without horizontally chiselled stones or clay bricks | 5 | Regular blocks made of soft stone (tuff, calcarenite etc.) |

**Table 1.** *Cont.*

| | Tuscany Region [71] | | Italian Code [29] |
|---|---|---|---|
| F | Rubble stone masonry of variable dimensions with horizontally chiselled stones or clay bricks | 6 | Stone square blocks |
| G | One-leaf masonry made by blocks of tuff of chiselled stones with constant dimensions | 7 | Clay bricks and lime mortar |
| H-I | Prefabricated concrete blocks with ordinary or light homogenous inserts | 8 | Semi-full bricks with cement mortar (ex. Double UNI with the hollow part ≤40%) |
| L | Full or semi-full clay masonry | | |
| M | Clay block masonry with the dimensions of the hole being greater than 45% | | |
| T | Mixed structure, i.e., a combination of one of more of the previous typologies | | |
| U | Confined masonry | | |
| V | Reinforced masonry | | |
| Z | Consolidated masonry (injection of mortar, reinforced concrete layers, etc.) | | |

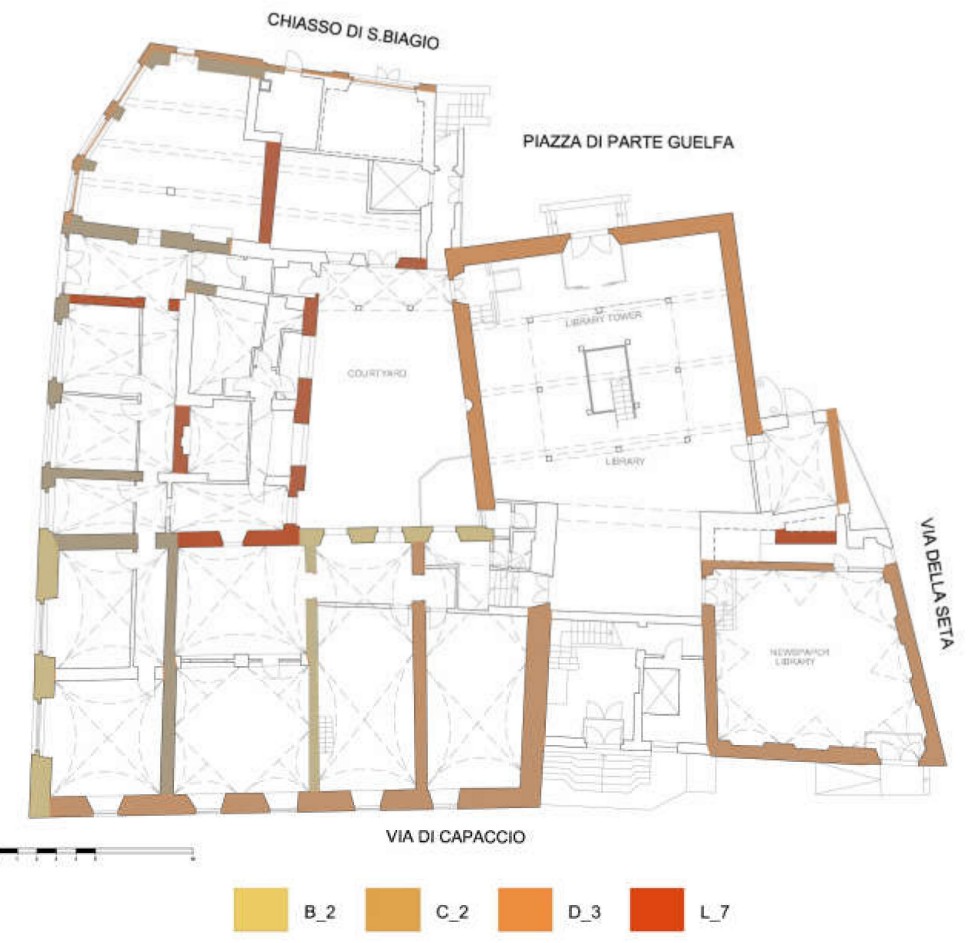

**Figure 8.** Layout of the building showing the identified typologies.

**Table 2.** Masonry typologies present in the Palagio di Parte Guelfa.

| | Tuscany Region [71] | | Italian Code [29] |
|---|---|---|---|
| B | Masonry comprising an inner nucleus and two external leaves realized with stones of regular dimensions, well-disposed and with connections between the two sides of the panel, horizontal layers of chiselled stone or clay bricks | 2 | Rough block with non-homogeneous thickness of the external faces |
| C | Rough stone masonry with irregularities | 2 | Rough block with non-homogeneous thickness of the external faces |
| D | Rough stone masonry with horizontal layers of chiselled stone or clay bricks | 3 | Split stones with good textural disposition |
| L | Full or semi-full clay masonry | 7 | Clay bricks and lime mortar masonry |

## 3. The Seismic Vulnerability Assessment

### 3.1. Definition of the Seismic Demand

The area around Florence is characterized by moderate seismic activity. The two main seismic sources are located along the vertical axis: one to the north in the Mugello area, and the other in the southern part at the beginning of the Chianti territory. The most significant historical earthquakes occurred in 1453 (MCS level around grade VII–VIII) and 1895 (grade VIII of the MCS scale) [73,74]. Considering the study site, the Italian code considers a peak ground acceleration PGA for a return period of 475 years equal to 0.131 g for a rigid soil or bedrock. In the new seismic micro-zoning of the city of Florence (in the public domain at https://emidius.mi.ingv.it/CPTI15-DBMI15/place/IT_45020 accessed on 5 November 2022), the site of the Palagio di Parte Guelfa has an amplification factor of between 1.97 and 2.39. In addition, the local seismic response of the area can be defined; this considers the geometric and seismic-stratigraphic information of the soils in order to define the in situ seismic design spectrum. The area of the Palagio is characterized by bedrock 15 m under the current level of the street. Over it, the soil stratigraphy consists of 5 m of archaeological layer and 10 m of sandy gravel. In an analysis of the soil stratigraphy in terms of shear-wave velocity, soil class B was identified, according to the prevision of the Italian Rule [75] (Figure 9a). The code spectrum for the reference area and an important class factor equal to III was adopted for the seismic analyses; for kinematic evaluations of the mechanisms located at different heights of the structure, the floor spectra formulation presented in [75] was used. The spectra vary from floor to floor according to the fundamental period of each structure, $T_0$, and the considered level of the analysis (z/H ratio). In Figure 9b, a set of spectra for the different structural units of Palagio di Parte Guelfa are reported. Considering the fundamental period, the simplified formulation provided by [76] for masonry structures was used ($T_0 = 0.05 \, xH^{0.75}$). Hence, for each unit, different spectra were considered based on the position of the hinges.

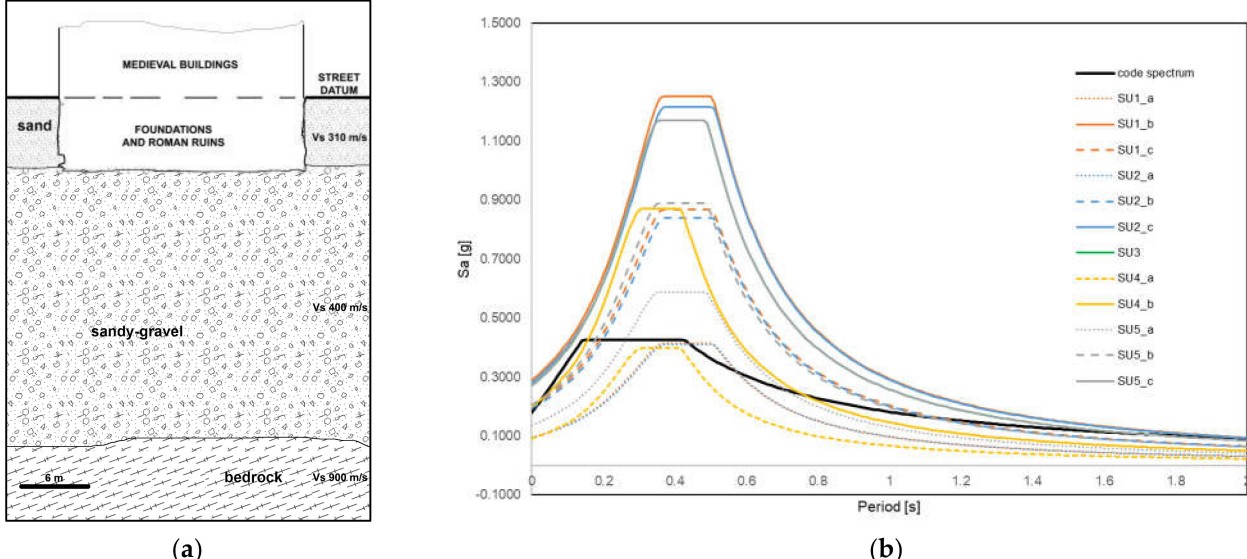

**(a)** **(b)**

**Figure 9.** (**a**) Information about the soil velocity for the different layers according to soil class; (**b**) considered floor spectra for the different structural units of the palace.

### 3.2. Vulnerability Assessment

In this work, the "Guidelines for the evaluation and reduction of seismic risk of cultural heritage" [14] were followed. As presented in the flowchart in Figure 2, the simplified level, LV1, was evaluated for all the structural units. Different models are available, depending on the structural and architectural features of the investigated buildings (Tower, Palace, Church, and Bridge). According to the characteristics of the Palagio, two distinct models were adopted: the Palace and Church models.

The Palace model adopts a simplified approach to determine the shear forces of each level considering the bearing walls. To determine the Life Safety Limit State (SLV), the ordinate of the elastic spectrum $S_{e,SLV}$ is determined as follows:

$$S_{e,SLV} = \frac{q\,F_{SLV}}{e^*M} \tag{1}$$

where $F_{SLV}$ represents the shear resistance of the bearing walls, q is the q-factor coefficient, M the total seismic mass, and e* is the fraction of participant mass over the first vibration mode. Once the ordinate of the response spectrum is computed, the return period $T_{SLV}$ for the corresponding seismic action can be defined. Collapse occurs when the tangential mean stress attains a given rate of the shear resistance of the masonry material:

$$F_{SLV,xi,yi} = \frac{\mu_{xi,yi}\xi_{xi,yi}\zeta_{x,y}A_{xi,yi}\tau_{di}}{\beta_{xi,yi}k_i} \tag{2}$$

where $A_{xi,yi}$ represents the resistant area of the i-th floor; τdi is the shear resistance of the masonry walls; $\mu_{xi,yi}$ is a coefficient considering the homogeneity of the stiffness and resistance of the bearing walls [14]; $\xi_{xi,yi}$ is related to the type of failure of the piers of the i-th level, i.e., 0.80 for flexural failure and 1.00 for shear; $k_i$ is the ratio of the seismic forces on the i-th floor and the total seismic force; and $\beta_{xi,yi}$ is a coefficient representing the planar irregularity at the i-th level, associated with the eccentricity between the barycenters of mass and stiffness of the system. The shear resistance was calculated on the basis of the masonry typologies identified in Table 2. The Italian Guidelines indicate that the computation of seismic acceleration has to be conducted for two directions and for the different levels. Later, the most vulnerable level and seismic direction indicate the final risk index value for the considered building.

In contrast, the Church model follows a macro-element approach, where the vulnerabilities of a structure are identified by means of the most likely outcomes during a seismic event. The procedure is recommended for churches and other structures where the masonry walls do not have intermediate connections (inter-storeys, slabs) which would guarantee a box behavior. The methodology considers 28 damage mechanisms, associated with the recurrent ones highlighted during the post-seismic surveys. For both methodologies, a final risk index can be determined. In the present work, the latter approach was used to obtain Safety Index Is, computed as the ratio of the seismic capacity to seismic demand (in terms of accelerations). This fraction indicates a value of Is which is equal to or bigger than 1. In any case, in existing structures (and specifically in relation to CHBs), an Is value bigger than 0.60 is still considered acceptable, due to the fact that these structures were not realized with consideration of seismic concepts. In Table 3, the safety indexes obtained from the simplified assessment for a SLV seismic action are shown. The results reveal differences between the two methodologies. In fact, the Church method is able to convert the vulnerability indexes into a risk index through an empirical formulation [14] that, although calibrated for use in Italy, is dimensionally related to possible mechanisms and does not depend upon the specific features of a particular structure. On the other hand, the risk evaluation provided by the Palace model accounts for effective loads and resistant areas along the two directions and for various masonry types, being more specific on the investigated structures. For the unit which was originally a church, the final safety index was satisfactory, as it exhibits a compact geometry with three macroelements, thus reducing its vulnerability value, which was found to be 0.40. The conversion formula adopted to transform the vulnerability index into a risk index later led to a $I_{s,SLD}$ equal to 0.60 and $I_{s,SLV}$ of 1.11. On the other hand, the other structures presented lower safety indexes, with a mean value of 0.31.

**Table 3.** Safety indexes for the different structural units SU comprising the Palagio di Parte Guelfa (the numbers of the different units are consistent with those used in Figure 6).

| S.U. | 1 | 2 | 3 | 4 | 5 | 6 |
|---|---|---|---|---|---|---|
| $I_{s,SLV}$ | 0.293 | 0.376 | 0.351 | 0.250 | 0.275 | 1.11 |

In addition to LV1, LV2 was also evaluated, based on a kinematic assessment of the most likely mechanisms of the different structural units. The kinematic approach was preferred to global assessments, as the knowledge path showed units without significant box behavior, where the big dimensions of the spaces, both in width and height, lead to independent parts that do not collectively resist seismic forces. For each mechanism, by applying the principle of virtual work, it is possible to compute the multiplier that activates the kinematic action. With both levels of evaluations, the Confidence Factors (CF) were, based on the obtained information, defined according to the suggestions of MIBACT.

For each structural unit, several out-of-plane mechanisms were identified considering several macroelements (capital letters, Figure 10). Based on the obtained data, the most likely kinetic action is the one activated by lower acceleration (leading to the low safety index). In Table 4, the results of the kinematic assessment are shown. Different mechanisms were identified, such as simple overturning (OT), composed overturning (COT), the vertical flexural qualities of the panels (VF), and overturning involving the top part (TOP) of the structure or the relative merlons. Plausible mechanisms were identified on the basis of the building's historical evolution and the accumulation of the urban aggregate. In case of masonry walls closing the inner space between pre-existing structural units, as the new facades are not orthogonally connected to the other bearing walls, these will be more vulnerable to overturning actions. The results for the first column $I_{s,SLV}$ excluded the contributions of the existing tie rods (worst condition). In column T of Table 4, the tension forces needed to prevent the activation of the mechanisms are shown. The blue boxes refer to the cases where tie rods were found, the other boxes indicate design forces for strengthening interventions. The tension loads were compatible with the dimensions and

number of elements in all the different units; however, in anticipation of a seismic event, other issues should be considered (e.g., the slipping of the tie rods in case of the absence of bolted end-plates). In Figure 11, a final summary of the minor indexes obtained for each structural unit is presented. The results are presented in different colors based on the safety level of each mechanism. Once again, although a safety index higher than 1 is desirable, for cultural heritage buildings, values bigger than 0.6 are considered acceptable, because such buildings cannot be reinforced if this would compromise their heritage value.

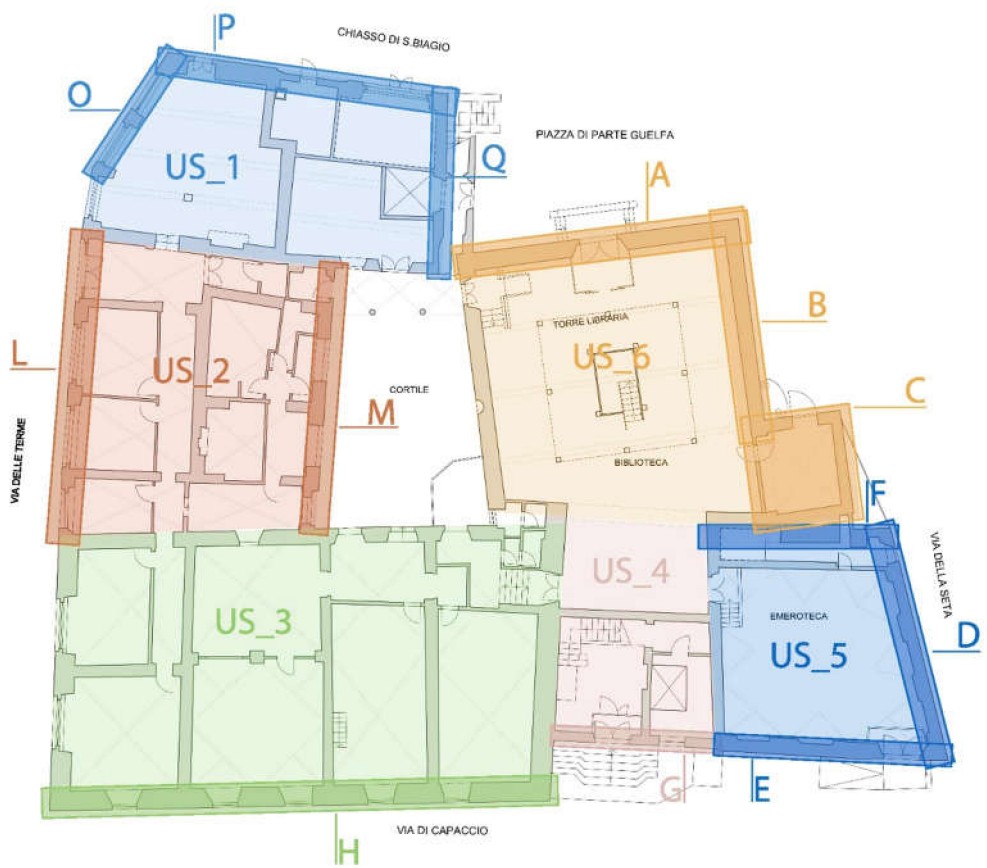

**Figure 10.** Identification of the considered macro-elements for the different structural units.

The values less than 0.6 are colored dark red, those between 0.60 and 0.99 in orange, and those greater than 1 in green. In general, the lowest indexes result from the existence of vulnerable and secondary elements at the highest level of the structure (e.g., medieval merlons) which are subjected to higher acceleration because of their positions and slender proportions with no stabilizing action. Mechanisms involving bigger portions of the structures are less likely to occur and have bigger Is values. As expected, simple overturning yields safety indexes below those resulting from the vertical flexural mechanism. Regarding some macroelements (for instance, macroelement M for US2), our kinematic analysis identified critical values, assuming the absence of a contribution of the tension of the existing tie rods (which is unknown); nonetheless, compatible tension values inside the steel bars would prevent out-of-plane effects due to ground motions. The lowest values were observed in structural unit n. 1 in both evaluations, i.e., LV1 and LV2; however, all the structural units presented critical values.

**Table 4.** Kinematic analyses applied in this research. In the column of the tie rods, the blue boxes refer to existing elements.

| Structural Units | Macro-Element | Mechanism | $I_{s,SLV}$ | T (kN) | $I_{s,SLV}$ |
|---|---|---|---|---|---|
| US_6 | A | OT1 | 0.6 | 80 | 1 |
| | C | COT side1 | 0.24 | 18.4 | 1.01 |
| | | COT side 2 | 0.4 | 18.4 | 1.01 |
| US_5 | D | VF 2+3 | 0.95 | 80 | 1.03 |
| | | COT 2+3 | 0.69 | 92 | 1.01 |
| | | COT 3 | 0.72 | 30 | 1.01 |
| | | TOP | 0.81 | 10 | 1.04 |
| | E | COT1+2+3 | 0.41 | 135 | 1 |
| | | COT 2+3 | 0.87 | 29 | 1.01 |
| | | COT 3 | 0.76 | 40 | 1.03 |
| | | VF 1+2 | 0.41 | 170 | 1.06 |
| | | VF 2+3 | 0.97 | 10 | 1.08 |
| | F | COT 3 | 0.71 | 24 | 1.02 |
| US_4 | G | COT1+2+3 | 0.32 | 50 | 1.11 |
| | | COT 2+3 | 0.42 | 35 | 1.08 |
| | | COT 3 | 0.13 | 12 | 1.04 |
| | | VF 1+2 | 1.06 | | |
| | | VF 2+3 | 0.54 | 35 | 1.02 |
| US_3 | H | OT | 0.59 | 200 | 1.03 |
| | | COT1+2+3 | 1.24 | | |
| | | COT2+3 | 1 | | |
| | | COT2+3/2 | 1.62 | | |
| | | COT3 | 1.07 | | |
| US_2 | L | COT1+2+3+4 | 0.31 | 174 | 1 |
| | | COT 2+3+4 | 0.52 | 75 | 1.01 |
| | | COT 3+4 | 0.33 | 159 | 1.01 |
| | | COT 4 | 0.59 | 50 | 1.06 |
| | | VF 1+2 | 1.12 | | |
| | | VF 2+3+4 | 0.98 | 10 | 1.02 |
| | M | COT1+2+3 | 0.24 | 150 | 1.03 |
| | | COT 2+3 | 0.52 | 96 | 1.01 |
| | | COT 3 | 0.42 | 90 | 1.04 |
| | | VF 1+2 | 0.29 | 90 | 1.08 |
| | | VF 2+3 | 0.96 | 15 | 1.04 |

**Table 4.** *Cont.*

| Structural Units | Macro-Element | Mechanism | $I_{s,SLV}$ | T (kN) | $I_{s,SLV}$ |
|---|---|---|---|---|---|
| US_1 | O | COT1+2+3 | 0.57 | 60 | 1.03 |
| | | COT 2+3 | 0.6 | 50 | 1.03 |
| | P | COT1+2+3 | 0.91 | 40 | 1.03 |
| | | COT 2+3 | 0.82 | 60 | 1.05 |
| | | COT 3 | 0.17 | | |
| | | Top Merlon | 1.18 | | |
| | Q | COT 2+3 | 0.8 | 30 | 1.01 |

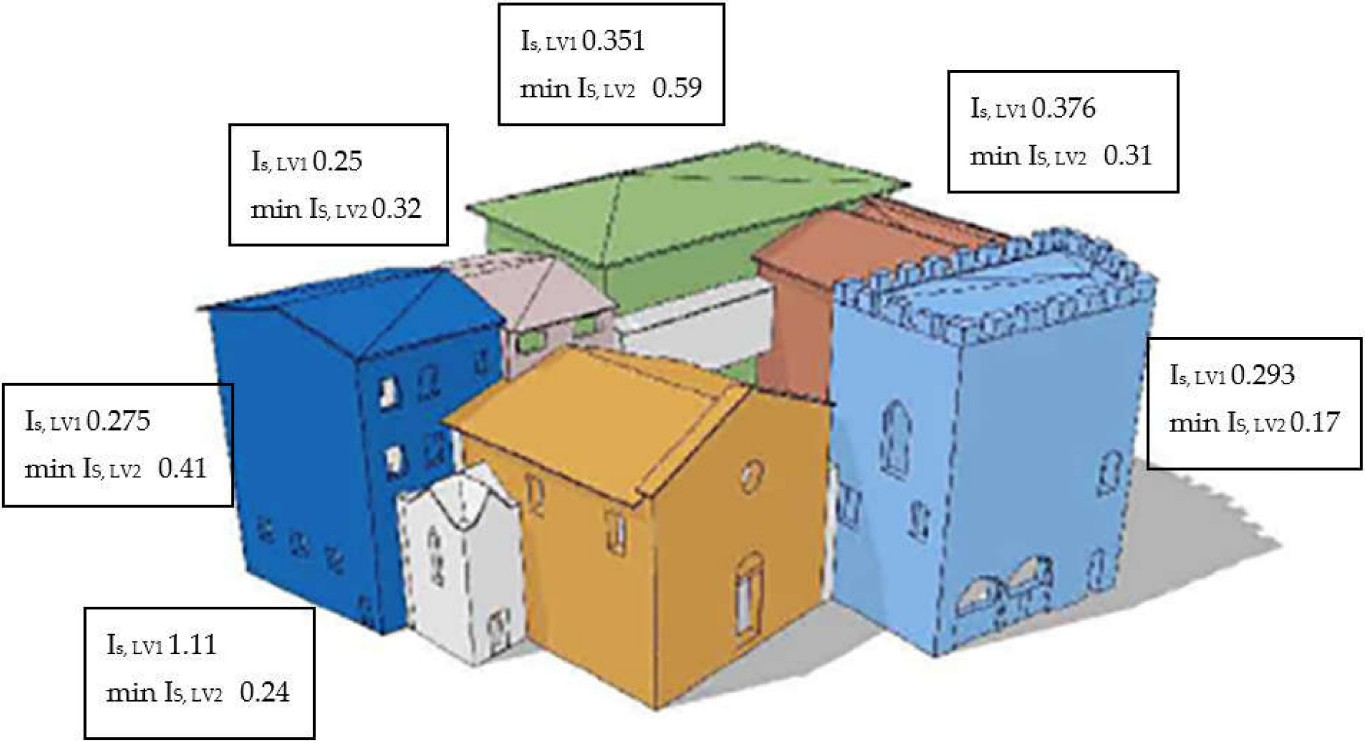

$I_{s,LV1}$ 0.351
min $I_{s,LV2}$ 0.59

$I_{s,LV1}$ 0.25
min $I_{s,LV2}$ 0.32

$I_{s,LV1}$ 0.376
min $I_{s,LV2}$ 0.31

$I_{s,LV1}$ 0.275
min $I_{s,LV2}$ 0.41

$I_{s,LV1}$ 0.293
min $I_{s,LV2}$ 0.17

$I_{s,LV1}$ 1.11
min $I_{s,LV2}$ 0.24

**Figure 11.** Summary of the safety indexes obtained for each structural unit of the palace.

## 4. Conclusions

In this paper, a seismic vulnerability assessment of the Palagio di Parte Guelfa in Florence has been presented. In this research, a holistic, multilevel procedure based on ND and MD techniques allowed us to investigate the seismic vulnerability of a complex urban aggregate identified as a heritage building. The methodology followed different steps, combined into a hierarchical workflow. Specifically, historical research and an integrated survey allowed us to define the structural units of the heritage cluster. ND and MD techniques were then adopted to characterize the structural system of each unit, providing evidence of the composition and quality of the bearing masonry walls, slabs, and vaults. The investigation provides general and specific outcomes. The first ones are related to the limited invasive procedures that the workflow required for the investigation of a complex CHB. The authors hope that the applied methodology will encourage new investigations of different CH structures in other national and international contexts. The specific outcomes of this study deal with the vulnerability of the Palagio di Parte Guelfa, an important palace in the historical center of Florence. Our assessment of the Palagio took advantage of a significate knowledge path which was elucidated through historical research, LS surveys, and ND techniques. The combination of the different in situ campaigns allowed us to identify the architectural and structural features of the building. The palace has been divided into different structural units and the seismic performance of each has

been independently investigated. Two levels of evaluations were executed by means of simplified approaches and kinematic analysis. The results show critical safety indexes based on several factors:

(i)    the Palace was not designed in accordance with seismic codes, but rather, based on empirical assumptions; the big dimensions of the buildings do not give rise to box behavior;

(ii)   the different macroelements tend to behave independently.

(iii)  the important class function of the palace increases the seismic demand

Further studies could be executed, with limited minor destructive/destructive campaigns to validate the findings presented in this work. Additionally, the influences of the aggregations and the adjacent structures on the seismic performance of the individual buildings could be evaluated in future research.

**Author Contributions:** Conceptualization, M.C., M.T. and M.D.S.; methodology, V.C., A.L.C. and M.T.; software, A.C. and E.L.P.; validation, A.L.C., V.C. and M.T.; investigation, A.C., E.L.P., V.C. and A.L.C.; resources, M.C. and M.D.S.; data curation, M.T.; writing—original draft preparation, V.C.; writing—review and editing, V.C. and M.C.; supervision, M.C. and M.T. All authors have read and agreed to the published version of the manuscript.

**Funding:** This research was funded by Comune di Firenze, grant number COL19COMFI1MONUMENTI2 and DESTEFANOSISMICAFIRENZE2.

**Institutional Review Board Statement:** Not applicable.

**Informed Consent Statement:** Not applicable.

**Data Availability Statement:** Not applicable.

**Acknowledgments:** This research belongs to the Protocol signed between the University of Florence and the Municipality of Florence for the seismic vulnerability assessment of monumental historical buildings. The authors acknowledge the work of F. Fazzari and R. Molinari conducted during their Master's thesis.

**Conflicts of Interest:** The authors declare no conflict of interest.

## Appendix A

Figure A1 refers to the investigations carried out during the experimental campaigns. Figure A2 shows the masonry typologies identified as the result of the different studies.

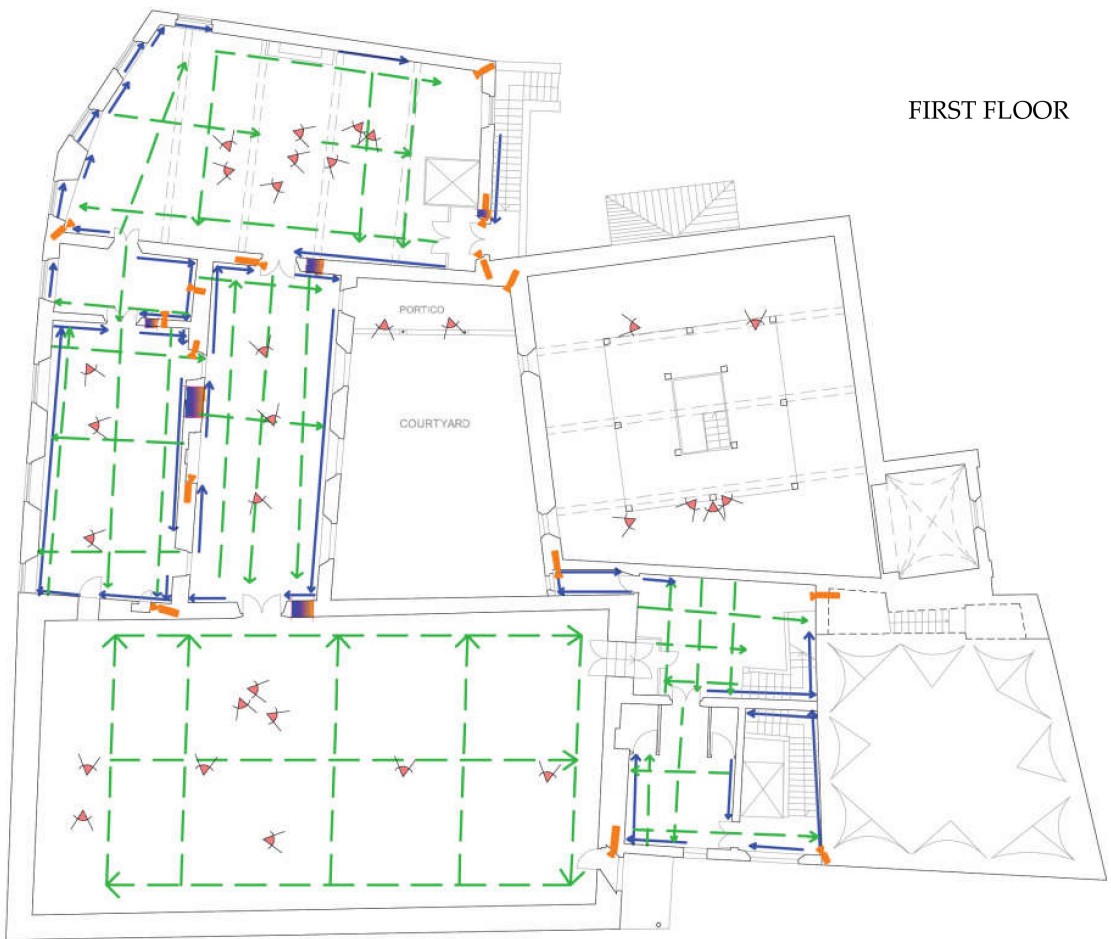

FIRST FLOOR

**Figure A1.** *Cont*.

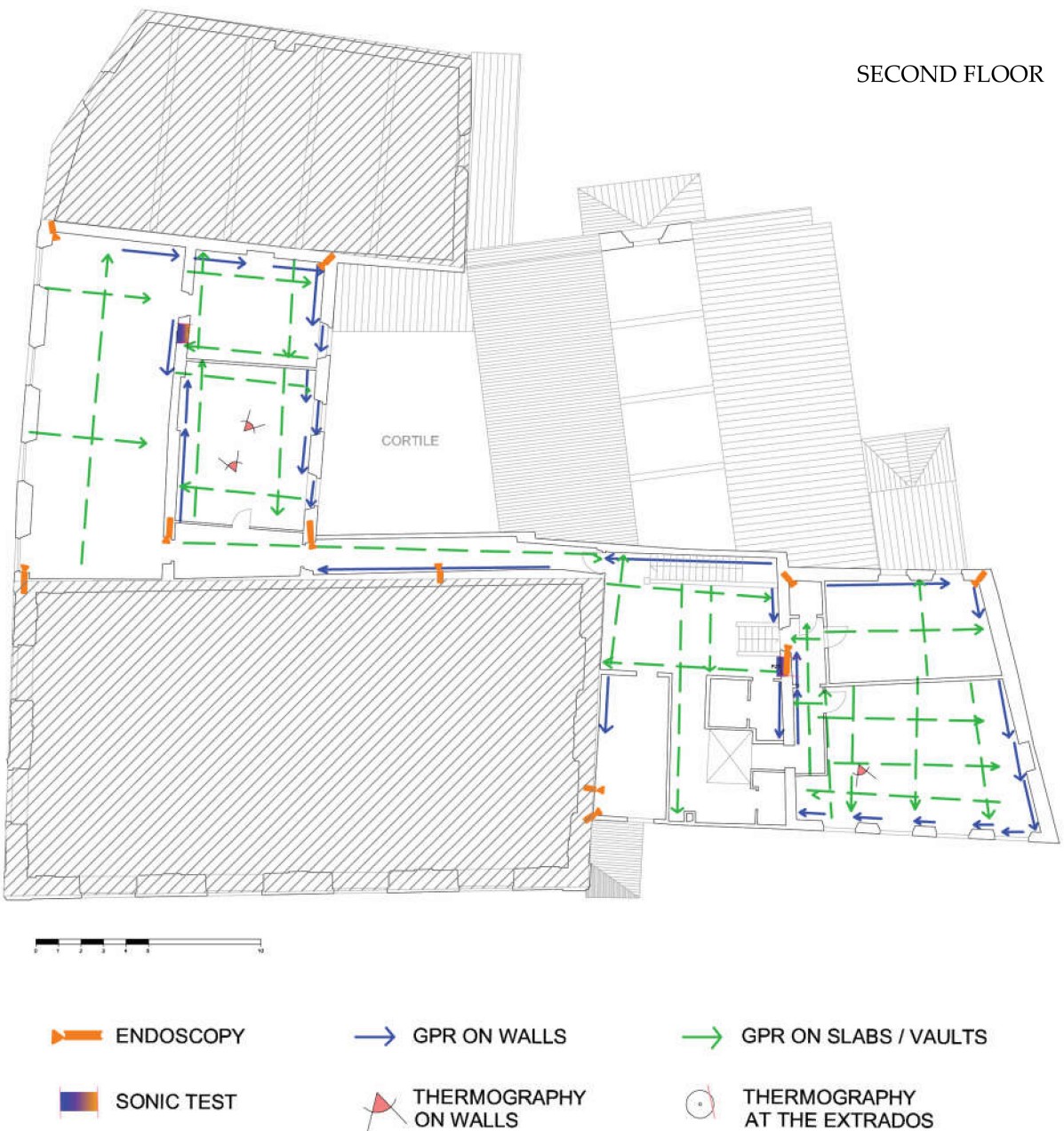

SECOND FLOOR

**Figure A1.** Investigations carried out on the different parts of the building.

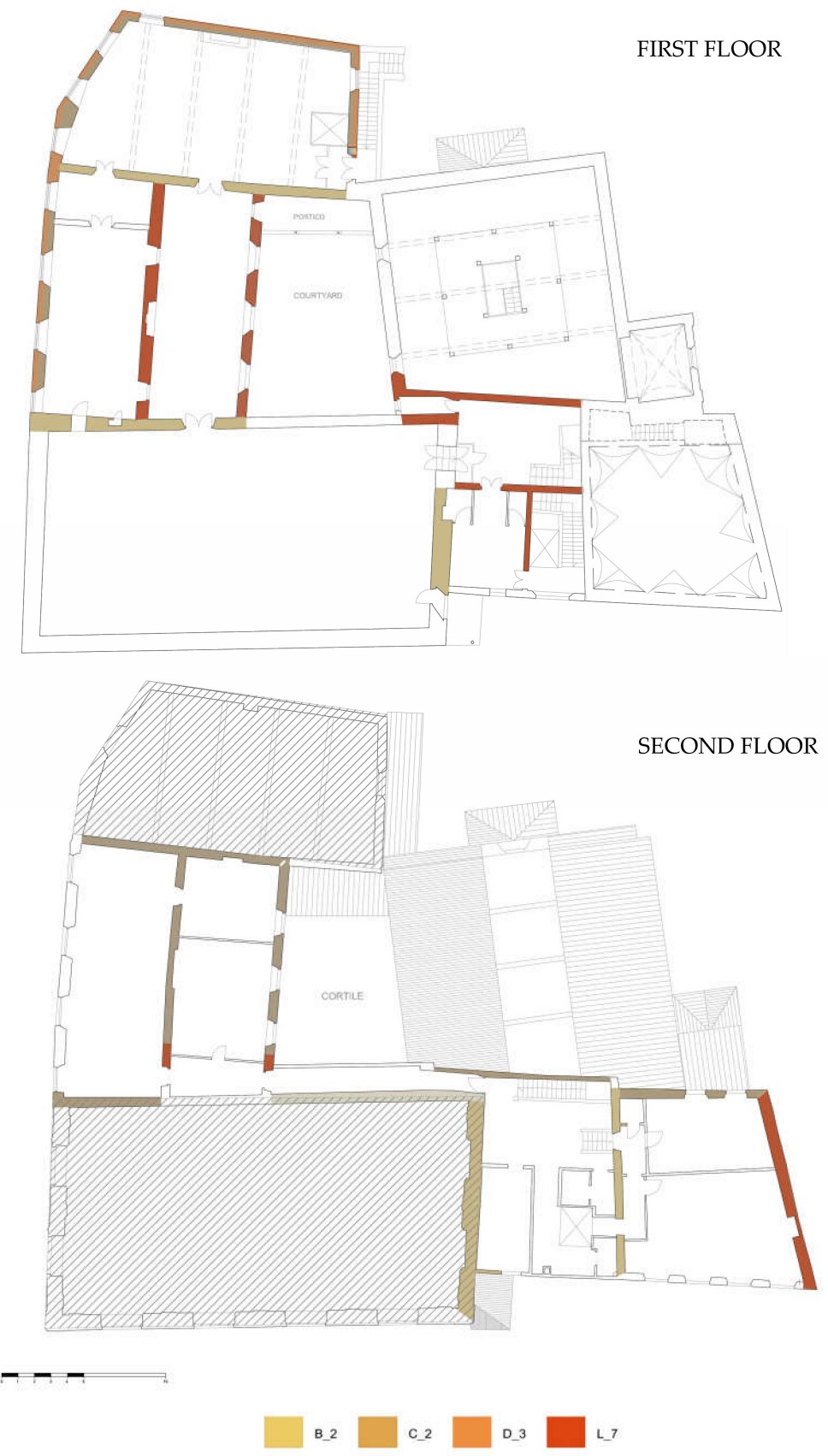

FIRST FLOOR

SECOND FLOOR

B_2    C_2    D_3    L_7

**Figure A2.** Identification of the masonry walls on the different levels.

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
