# Peer review of "Seismic Vulnerability Assessment of a Medieval Urban Cluster Identified as a Complex Historical Palace: Palagio di Parte Guelfa in Florence"

_heritage, doi:10.3390/heritage5040217_

Round 1

Reviewer 1 Report

The manuscript provides an interesting investigation of a complex historical masonry Palace, based on historical research, in-situ inspections, experimental tests and seismic analyses. The subject is worthy of consideration, the paper is well conceived and organized. The manuscript can be accepted after revisions, based on the following suggestions provided to the Authors.

1) Abstract should be improved, better highlighting the novelty of the work. It is suggested to add the main novel result (in a very synthetic way) obtained in the study.

2) Section 1 (Introduction). A more comprehensive explanation of the main novelty aspects of the study should be provided at the end of Introduction.

3) The authors can mention (near references [21-24]) the following reference, which presents an advanced approach based on kinematic limit analysis for local failure mechanisms evaluation of historical masonry aggregates:

https://doi.org/10.1007/s10518-020-00848-6

4) It is suggested to improve, if possible, the quality and clarity of the plans shown in Figures 5, 6, 7.

5) Line 323-328. It is suggested to provide more information about the analysis of the soil stratigraphy, the code spectrum and the floor spectra adopted.

6) Section 3.2, Lines 336-7. The simplified approach adopted (Palace model) should be better described and explained.

7) Section 3.2, Line 344. A more comprehensive description and explanation of the Safety Index should be provided.

8) Line 347-354. Better explain the “differences within the two simplified methodologies”.

9) The results presented in Table 4 should be better discussed.

10) Conclusions should be improved and enlarged, better summarizing the main findings obtained in this study. Moreover, the main novelty aspects of the study should be better highlighted.

11) The manuscript is generally well-written: however, a general revision of the text is required.

Author Response

Reviewer 1

The manuscript provides an interesting investigation of a complex historical masonry Palace, based on historical research, in-situ inspections, experimental tests and seismic analyses. The subject is worthy of consideration, the paper is well conceived and organized. The manuscript can be accepted after revisions, based on the following suggestions provided to the Authors.

  1. Abstract should be improved, better highlighting the novelty of the work. It is suggested to add the main novel result (in a very synthetic way) obtained in the study.

The abstract has been improved highlighting the novelty of the methodological approach adopted to investigate complex masonry aggregates ascribable to cultural heritage palaces.

  1. Section 1 (Introduction). A more comprehensive explanation of the main novelty aspects of the study should be provided at the end of Introduction.

The following sentences has been added to the text. In addition, further comprehensive explanations of the main novelty aspects of the research have been denounced in the conclusions.

“In this work, a methodological framework combining historical research, integrated architectural and structural survey with ND and MD techniques has been adopted. The approach showed the feasibility of the procedure to investigate complex CHBs, although limiting the side-effects required by the knowledge path.”

  1. The authors can mention (near references [21-24]) the following reference, which presents an advanced approach based on kinematic limit analysis for local failure mechanisms evaluation of historical masonry aggregates: https://doi.org/10.1007/s10518-020-00848-6

The suggested paper has been added to the list of references

  1. It is suggested to improve, if possible, the quality and clarity of the plans shown in Figures 5, 6, 7.

The figures have been improved. More emphasis has been given to the ground floor plan only, while the other floors have been put in Appendix A.

  1. Line 323-328. It is suggested to provide more information about the analysis of the soil stratigraphy, the code spectrum and the floor spectra adopted.

In Fig. 9 more information on the soil stratigraphy and the soil spectra have been provided. In addition, the following lines have been added to the text:

“The spectra vary floor by floor as a function of the fundamental period of each structure T0 and the considered level of the analysis (z/H ratio). In Fig. 9b a set of spectra for the different structural units of Palagio di Parte Guelfa are reported. Considering the fundamental period, the simplified formulation provided by [NTC2008] for masonry structures has been used (T0=0.05*H0.75). Hence, for each unit different spectra are considered based on the position of the hinges.”

  1. Section 3.2, Lines 336-7. The simplified approach adopted (Palace model) should be better described and explained.

The palace model has been described more in detail in the manuscript. The following part has been added to the text:

 “For the achievement of the Life Safety Limit State (SLV), the ordinate of the elastic spectrum Se,SLV is accounted as:

                                                                                                                       (1)

Where FSLV represents the shear resistance of the bearing walls, q is the q-factor coefficient, M the total seismic mass, e* the fraction of participant mass over the first vibration mode. Once the ordinate of the response spectrum is computed, the return period TSLV for the correspondent seismic action can be defined. The collapse is reached when the tangential mean stress attains a given rate of the shear resistance of the masonry material:

                                                                                                     (2)

With Axi,yi representing the resistant area of the i-th floor, ?di the shear resistance of the masonry walls.  is a coefficient considering the homogeneity of stiffness and resistance of the bearing walls [Mibact2011], while  is related to the type of failure of the piers of the i-th level, accounted as 0.80 for flexural failure and 1.00 for shear. ki is the ratio within the seismic forces at the i-th floor and the total seismic force. βxi,yi is a coefficient representing the planar irregularity at the i-th level, associated with the eccentricity between the barycenters of mass and stiffness of the system. The shear resistance has been accounted on the basis of the masonry typologies identified in Table 2. The Italian Guidelines specifies the computation of the seismic acceleration has to be conducted according to the two directions and for the different levels. Later, the most vulnerable level and seismic direction indicates the final value of the risk index for the considered building.”

  1. Section 3.2, Line 344. A more comprehensive description and explanation of the Safety Index should be provided.

A more comprehensive description and explanation has been provided. The sentence has been implemented as it follows:

“In the present work, the latter is accounted as a Safety Index Is, computed as a ratio within the seismic capacity and the seismic demand (both in terms of accelerations). This fraction indicates the computations as verified for Is equal or bigger than 1. In any case, investigating existing structures (and specifically in relation to CHBs) Is values bigger than 0.60 are still considered acceptable, due to the fact that these structures have not been realized following seismic concepts.”

  1. Line 347-354. Better explain the “differences within the two simplified methodologies”.

The text has been rephrased. The following part is now written in the manuscript:

“The results point out differences within the two simplified methodologies. In fact, the Church method allows to convert the vulnerability indexes into a risk index through an empirical formulation [14] that, although calibrated on the Italian territory, is a-dimensionally related to the possible mechanisms, without involving the building-by-building features of the investigated structure. On the other side, the risk computation provided by the Palace model, as it accounts the effective loads, resistant areas along the two directions and masonry types, is more calibrated on the specific investigated buildings.

  1. The results presented in Table 4 should be better discussed.

The results have been discussed more in detail

  1. Conclusions should be improved and enlarged, better summarizing the main findings obtained in this study. Moreover, the main novelty aspects of the study should be better highlighted.

The conclusions of the work have been reviewed according to the suggestion of the reviewer

  1. The manuscript is generally well-written: however, a general revision of the text is required.

A general revision of the text has been done

Reviewer 2 Report

The paper presents the seismic vulnerability assessment of a significant Palace located in the historical centre of Florence, named the Palagio di Parte Guelfa.

The paper is well structured, describing in a sufficient manner the issue investigated. Therefore, it is opinion of the Reviewer that the paper deserves to be published in the Journal. However, a first round of revision should be performed, accounting for the following comments.

Point 1. There are a lot of recent works addressed to the application of the multilevel approach for seismic assessment of cultural heritage according to the Italian Guidelines. Among the others the following additional works may be referenced, where also a discussion of the simplified models proposed by the Italian design codes are discussed in depth:

·      Formisano A, Marzo A. Simplified and refined methods for seis- mic vulnerability assessment and retrofitting of an Italian cultural heritage masonry building. Comput Struct. 2017;180:13–26.

·      D'Amato, M., Sulla, R., 2021. Investigations of masonry churches seismic performance with numerical models: application to a case study. Archives of Civil and Mechanical Engineering 21, 161. https://doi.org/10.1007/s43452-021-00312-5

Point 2. Please be sure that all the acronyms are explained within the text.

Point 3. As depicted in the Flow-chart of Fig.2, after preliminary investigations, in situ-inspections and historical researches have been conducted on the case study. Reviewer does not understand what is intended for ‘preliminary investigations’. 

In addition, the type of in-situ inspections carried-out in this case should be justified, in order to better understand the importance of the results achieved in the knowledge path of the case study. Please comment on this.

Point 4. The quality of some figures does not seem sufficient. Please check all figures. As for Figure 4 and Figure 5, please add a legend explaining the two colors used in the drawings.

Point 5. Reviewer really appreciates the historical evolution investigated for the case study analyzed. However, it is not clear how the evolution of structural configuration has been considered in the numerical models for seismic assessment of the case study. Therefore, Authors are invited in clarifying this issue.

Author Response

Reviewer 2

The paper presents the seismic vulnerability assessment of a significant Palace located in the historical centre of Florence, named the Palagio di Parte Guelfa. The paper is well structured, describing in a sufficient manner the issue investigated. Therefore, it is opinion of the Reviewer that the paper deserves to be published in the Journal. However, a first round of revision should be performed, accounting for the following comments.

  1. Point 1. There are a lot of recent works addressed to the application of the multilevel approach for seismic assessment of cultural heritage according to the Italian Guidelines. Among the others the following additional works may be referenced, where also a discussion of the simplified models proposed by the Italian design codes are discussed in depth:

Formisano A, Marzo A. Simplified and refined methods for seis- mic vulnerability assessment and retrofitting of an Italian cultural heritage masonry building. Comput Struct. 2017;180:13–26.

D'Amato, M., Sulla, R., 2021. Investigations of masonry churches seismic performance with numerical models: application to a case study. Archives of Civil and Mechanical Engineering 21, 161. https://doi.org/10.1007/s43452-021-00312-5

The references have been added to the paper

  1. Point Please be sure that all the acronyms are explained within the text.

We provided a revision of the acronyms in the text

  1. Point 3. As depicted in the Flow-chart of Fig.2, after preliminary investigations, in situ-inspections and historical researches have been conducted on the case study. Reviewer does not understand what is intended for ‘preliminary investigations’. 

In the text, referring to the preliminary investigation, the following sentences have been added:

“This phase anticipates the execution of accurate geometrical surveys or the application of diagnostic techniques. Conversely, it is targeted at getting confidence with the structure in view of the setup of the following steps. During these preparatory evaluations, preliminary structural models can be also realized, on the basis of the available information. They can be useful to execute sensitivity analysis highlighting the uncertain parameters (or given portions of the structure) that are worthy to be further investigated [Haddad et al. 2019]. In this work, before the specific investigation of the building, historical research has been made, acquiring the chronicles published by different authors.”

In addition, the type of in-situ inspections carried-out in this case should be justified, in order to better understand the importance of the results achieved in the knowledge path of the case study. Please comment on this.

All the investigations listed in the figure have been executed in the present research. The following clarification has been added to the text:

“In this research, all the different investigations shown in Fig. 2 have been adopted. Namely, architectural and structural survey of the building has executed. Then, sever-al ND and MD techniques have been performed, such as thermography campaigns, GPR surveys, sonic test on the masonry walls, drilling inspections.”

  1. Point 4. The quality of some figures does not seem sufficient. Please check all figures. As for Figure 4 and Figure 5, please add a legend explaining the two colors used in the drawings.

The quality of Fig. 6-7-8 has been improved. For Fig. 4 and Fig. 5, an explanation of the colors was added in the legends:

Figure 4. Plan of the first floor: hypothesis on the transformations in the years 1415-1426. In grey, the existing structures at the time are reported. In red, the possible new edifications are highlighted.

Figure 5. Resume of the transformations occurred from 1270 to 1923. For each phase, the red parts indicate the new volumes realized during those years.

  1. Point 5. Reviewer really appreciates the historical evolution investigated for the case study analyzed. However, it is not clear how the evolution of structural configuration has been considered in the numerical models for seismic assessment of the case study. Therefore, Authors are invited in clarifying this issue.

The study of the historical evolution of the structure has been important in order to identify: i) the structural units composing the complex; ii) determine possible mechanisms on the basis of the geometrical dis-connections due to the evolution of the Palace. Regarding the first point, the historical research helped in the comprehension of the current state of the building and the relationships between the different parts. For the second point, It is worth noting that when an area is saturated by new additions between distinct an independent structures, generally the new volumes are realized only erecting two new bearing walls, taking advantages of the adjacent buildings. This condition leads to un-connected structures, where the out-of-plane mechanisms express a higher potentiality. In the text the following sentence has been added:

“The plausible mechanisms have been supposed on the basis of the historical evolution and the saturation of the urban aggregate. In case of masonry walls closing the inner space between pre-existing structural units, as the new facades are not orthogonally connected to the other bearing walls, these will be more vulnerable towards overturning actions.”

Round 2

Reviewer 1 Report

The authors have addressed the concerns raised by the reviewer.

The revised manuscript is recommended for publication.

Reviewer 2 Report

The paper has been revised according to the comments provided. It is opinion of the Reviewer that it may be considered for the publication in the current form.